Subject Areas:
ecology/environmental science

Keywords:
habitat restoration, land use change, Markov chain, multi-layer perceptron, broadleaf expansion

Author for correspondence:
Syed Amir Manzoor
e-mail: s.a.manzoor@pgr.reading.ac.uk

# Scenario-led modelling of broadleaf forest expansion in Wales

Syed Amir Manzoor[1,2], Geoffrey Griffiths[3], James Latham[4] and Martin Lukac[1,5]

[1]School of Agriculture, Policy and Development, University of Reading, Reading, UK
[2]Department of Forestry and Range Management, Bahauddin Zakariya University, Bosan Road, Multan, Pakistan
[3]Department of Geography and Environmental Sciences, University of Reading, Reading, UK
[4]Natural Resources Wales, Wales, UK
[5]Faculty of Forestry and Wood Sciences, Czech University of Life Sciences Prague, Prague, Czech Republic

(iD) SAM, 0000-0002-2203-4696

Significant changes in the composition and extent of the UK forest cover are likely to take place in the coming decades. Current policy targets an increase in forest area, for example, the Welsh Government aims for forest expansion by 2030, and a purposeful shift from non-native conifers to broadleaved tree species, as identified by the UK Forestry Standard Guidelines on Biodiversity. Using the example of Wales, we aim to generate an evidence-based projection of the impact of contrasting policy scenarios on the state of forests in the near future, with the view of stimulating debate and aiding decisions concerning plausible outcomes of different policies. We quantified changes in different land use and land cover (LULC) classes in Wales between 2007 and 2015 and used a multi-layer perceptron–Markov chain ensemble modelling approach to project the state of Welsh forests in 2030 under the current and an alternative policy scenario. The current level of expansion and restoration of broadleaf forest in Wales is sufficient to deliver on existing policy goals. We also show effects of a more ambitious afforestation policy on the Welsh landscape. In a key finding, the highest intensity of broadleaf expansion is likely to shift from southeastern to more central areas of Wales. The study identifies the key predictors of LULC change in Wales. High-resolution future land cover simulation maps using these predictors offer an evidence-based tool for forest managers and government officials to test the effects of existing and alternative policy scenarios.

## 1. Introduction

Global forest cover has receded rapidly in the recent past, largely due to the conversion of forests into pasture and croplands [1].

In contrast to many other parts of the world where forest cover is still declining, Europe has reversed this deforestation trend and forest cover here is increasing as a consequence of land abandonment and subsequent secondary succession [2] and as a result of deliberate planting [3]. Afforestation of Great Britain is a good example of such reversal: forest cover stood at 5% in the 1920s but increased to 13% by 2013 [4]. Early twentieth-century forestry policy in Britain focused on planting fast-growing non-native conifer tree species to boost domestic timber supply [5], nearly tripling forest cover in the process. However, extensive plantations of single-species non-native conifers resulted in the loss of important habitats [6,7], with a direct negative effect on species diversity [8]. In the latter part of the twentieth century, British forestry policy gradually changed to address a wider set of objectives to complement timber production, resulting in a broad focus on the expansion of native tree species cover [5].

At present, non-native conifers constitute 36% of the total forest area in Britain. Typically represented by Sitka spruce (*Picea sitchensis* Bong.), the conversion of these woodlands to native species, where appropriate, is an aspiration stated in the UK Forestry Standard Guidelines on Biodiversity (UKFS) [9]. Non-native plantations often consist of even-aged and often single-species conifers mostly present in the uplands. UKFS guidelines on biodiversity encourage the large-scale expansion of native woodlands, primarily by replacing non-native species by native broadleaves or Scots pine (*Pinus sylvestris* L.). 'Forest conversion' in this context thus refers to the silvicultural process of changing forest stands dominated by non-native conifers into forests composed of native tree species [9].

There are several reasons why this type of forest conversion should become an important tool of forestry policy in Britain. When compared to a conifer plantation, broadleaf tree species tend to increase soil pH [10], improve growth condition for ground vegetation and soil fauna [11–13], enhance nesting opportunities and seasonal availability of food to bird communities [14] and decrease the prevalence of insect pests [15] and plant pathogens [16]. Further, and no less important in densely populated landscapes, forest conversion to native broadleaves impacts on landscape aesthetics—an attribute keenly perceived and appreciated by the public [17]. The conversion of conifer stands to broadleaf woodland is also supported by forestry policy in Wales, specifically the rules governing the restoration of planted ancient woodland sites (PAWS) that have historically been afforested with conifers [18]. This strategy is also supported by the objectives of the UK Biodiversity Action Plan.

There is strong political interest to enhance forest cover of the UK (e.g. [19]), with the Welsh Government committed to increasing the overall forest area in Wales as its contribution to sustainable development. In 2010, the Welsh Assembly Government's Climate Change Strategy [20] recommended that woodland establishment rates be increased to 5000 ha per year for 20 years as an option for meeting Wales' carbon emission reduction targets. This figure was subsequently adopted by Welsh Government as a policy target in the form of planting 100 000 ha of new, primarily broadleaved, woodland by 2030. However, actual planting levels over the following years were insufficient to meet this. The Welsh Government then regressively reduced the target, first to 50 000 ha, and eventually to the current commitment to plant at least 2000 ha of woodland a year between 2020 and 2030 [19]. Despite these reductions in aspiration, the broad commitment to increasing woodland cover has been bolstered by recent legislation and policy development. The Environment (Wales) Act (2016) requires the Welsh Government's principal advisor on the environment and natural resources, Natural Resources Wales, to periodically produce a State of Natural Resources Report that makes an assessment of the state of natural resources in Wales and the extent to which they are sustainably managed. The first such report highlighted the need for the increased woodland cover to deliver multiple benefits [21]. In this context, maintaining a balance between conservation and sustainable development is seen as a challenge in policy making and landscape management in Wales.

Where possible, forest management policies should be evidence-based [22,23]. One tool that can be used to generate evidence and test policy scenarios is landscape modelling which can project the spatial and temporal implications of policies under consideration [24]. Spatially explicit land use and land cover (LULC) models can project land use change patterns according to given policy objectives and generate alternative scenarios [25,26], thus establishing a link between policy and implications on the ground [27]. The LULC change models offer an evidence-based approach to visualize, analyse and quantify LULC changes in *what-if* future scenarios, thus providing useful insights to policy makers and relevant stakeholders to set priorities and reasonable goals for sustainable forest management [28].

To date, several spatially explicit modelling environments have been developed by incorporating remote sensing and geographical information system tools to project future LULC change scenarios. Some of the most widely used models are based on Markov chain, logistic regression, artificial neural network and cellular automata models [29]. Integrated use of these models is often advised to overcome limitations of individual models and generate robust LULC change simulations [30]. In this study, we employed an

integrated multi-layer perceptron–Markov chain analysis (MLP–MCA) method to quantify historic LULC change and to model future scenarios of LULC change in Wales. The MLP–MCA is a robust and well-documented approach for modelling spatial and temporal LULC changes [31–33]. To develop realistic future scenarios of LULC change, it is critical to understand the spatial patterns of current LULC and to develop an understanding of the potential impacts of current and future policies affecting LULC change and, specifically, native woodland expansion. A key assumption underlying most LULC modelling is that socio-economic drivers of change remain stable over time; this is, however, unlikely in reality [34]. To meet this challenge, we considered the following key questions prior to embarking on the modelling exercise:

1. Under recent policies, what would the state (type and area) of forests in Wales be in 2030?
2. What are the implications of an alternative policy scenario designed to stimulate new woodland creation while considering other objectives, such as the conservation value of non-woodland habitats?

Thus, this paper presents an analysis of the current situation and likely future trends, together with an indication of policy requirements necessary to achieve the stated afforestation goal for Wales. We mapped historical patterns of LULC and LULC change in Wales and developed two contrasting future LULC scenarios based on (i) current trends and (ii) an alternative policy. We also discuss the usefulness of the resulting future LULC maps of Wales for habitat, biodiversity and ecosystem services analysis.

# 2. Methodology

## 2.1. Study area

Wales is a country with an area of nearly 21 000 km$^2$ and a population of over 3 million, most of which live in rural communities [35]. The population is unequally distributed, with most people living in coastal areas in the northeast and south Wales. The country is characterized by a wide variety of landscapes, reflecting both its rugged topography and a long history of agricultural settlement and industrialization. Significant areas of land (approx. 6000 km$^2$) are at an altitude above 300 m. Welsh countryside contains a range of important habitats, including woodlands, semi-natural grasslands, heathland, fens, bogs, coastal ecosystems including sand dunes and saltmarshes, and a diverse range of upland and montane habitats [6,36]. Only a small proportion—6%—of the country is occupied by arable agriculture, while the major land use types are grazing (77%) and forestry (15%) [37–39].

## 2.2. Modelling methodology

We made use of TerrSet Geospatial Monitoring and Modelling System (version 18.31, Clark Labs, Clark University, USA) [40]. Specifically, we used the 'Land Change Modeler' (LCM) tool in TerrSet to generate two future LULC change scenarios: business-as-usual and an alternative scenario.

### 2.2.1. Analysis of past change

The first step of this type of analysis generates the spatial pattern of changes that are discernible from the comparison of historical LULC maps. A minimum of two maps is required (describing two different points in time), the comparison seeking to understand the nature of LULC change and to generate samples of transitions to be projected [40]. We used high-resolution (25 m) LULC maps of the country from 2007 and 2015, generated from satellite imagery by the Centre for Ecology & Hydrology, UK (https://digimap.edina.ac.uk) [41]. Both LULC maps had the same legend and spatial resolution. By comparing the two maps, LCM evaluates LULC change and generates a visual representation of net change, persistence, losses and gains and transitions between different LULC categories covered by the two raster maps. In this study, we excluded LULC categories that showed a negligible transition between 2007 and 2015 and thus are not expected to change over the time horizon under consideration (e.g. urban areas) and those not relevant to our objectives (e.g. water bodies or coastal areas). The final list of LULC categories selected for our modelling exercise is presented in the electronic supplementary material, table S1. We then generated change analysis maps indicating which LULC classes changed and the spatial pattern of changes across the Welsh landscape. This analysis of LULC change in Wales between 2007 and 2015 resulted in a total of 28 transition types between different LULC classes (electronic supplementary material, table S1).

## 2.2.2. Explanatory variables

### 2.2.2.1. Rationale for the choice of explanatory variables

Land use change modelling is complex, and a wide range of factors is likely to affect future decisions of landowners [42]. Therefore, the kind and number of explanatory factors for future land use change can vary considerably. Spatial and temporal scale of the study also affects the choice of explanatory variables [42]. Given that the spatial focus of this study is regional (i.e. Wales) and temporal scale is only 15 years (2015–2030), we applied the following rationale to the choice of explanatory variables:

— Biophysical variables. Biophysical factors such as topography or soil type influence land use and allocation decisions. For example, expansion of arable land may be limited by slope incline and/or altitude, planting new woodlands could be driven by soil quality or land parcel accessibility. Biophysical variables, especially at a fine spatial resolution of 25 m, are very strong proxies of climatic variables which are otherwise only available at a spatial resolution of 1 km or higher and thus cannot be used in fine-scale land use change modelling (e.g. altitude is a strong proxy for temperature, wind speed, etc.) [42,43].
— Proximate variables. At a regional scale, variables such as distance to markets or roads are strong determinants of landowners' decision-making [42,44]. For example, areas closer to roads or green spaces often have a higher market price. Areas in close proximities to major road junctions are more likely to experience change [45]. Furthermore, we used variables such as distance to exiting the broadleaf forest and arable land because the expansion of a habitat is very likely to be in the near vicinity of the already existing patches of that habitat [24,46].
— Evidence likelihood of land use transition. We mapped transitions from all land cover classes to the broadleaf forest and vice versa during 2007–2015 and used the evidence likelihood function to convert the patterns of these transitions into usable continuous variables. Evidence likelihood is an empirical probability of change for a qualitative map [47] and describes the relative frequency with which different LULC categories occurred within areas that transitioned between 2007 and 2015. These variables thus represent the likelihood of finding a specific LULC at the pixel in question, if the pixel covers an area suitable for the transition. Since the decision of change on a land parcel is strongly influenced by the decisions of the neighbouring land parcel [48–50], these variables can have important information.

### 2.2.2.2. Explanatory variables used in the study

As shown in table 1, we considered 20 variables in total as having the potential to explain LULC transitions occurring in Wales between 2007 and 2015. We collected a range of explanatory variables such as topography, soil factors or distance from key biophysical features such as water bodies and existing forests [33,51,52]. Six variables—distances between each of the six LULC classes—were dynamic, while the rest were static. Values of a dynamic variable change at each time step of the model run and thus need to be recalculated (e.g. the distance from the broadleaf forest as these forests expand). By contrast, static variables remain constant over time (e.g. altitude, slope, soil type).

Furthermore, we used the 'Evidence Likelihood Transformation' tool in the LCM to convert categorical variables to continuous (Clark Labs, 2015). The following transformations were made:

(1) We generated two Boolean images: first, change from all LULC types to Broadleaf Forest (All to Broadleaf) and change from Broadleaf Forest to all other LULC types (Broadleaf to All). In these images, 0 represents no change while 1 represents the indicated change. The Boolean images were then used in the Evidence Likelihood Transformation process to generate two continuous variables: 'Evidence Likelihood of change from Broadleaf Forest to All Other Classes' and 'Evidence Likelihood of change from All Other Classes to Broadleaf Forest'.
(2) Welsh soil type data used in this research consisted of 27 soil classes. We used the 'Evidence Likelihood Transformation' option to generate the following two continuous variables from this categorical variable: 'Evidence Likelihood of change from Broadleaf Forest to All Other Classes based on Soil type' and 'Evidence Likelihood of change from All Other Classes to Broadleaf Forest based on Soil type'.

The potential of each of the 20 variables to explain observed LULC change was tested by calculating Cramer's V [40]. Six variables were dropped from the final list on the basis of low Cramer's $V$ value ($V < 0.15$) [52].

**Table 1.** List of explanatory variables considered in the study. Variables in italics were chosen in the final model based on Cramer's *V* values ($V > 0.15$).

|  | explanatory variables | Cramer's *V* | type of variable |
|---|---|---|---|
| 1 | *altitude* | *0.345* | *static* |
| 2 | aspect | 0.077 | static |
| 3 | *slope* | *0.168* | *static* |
| 4 | hillshade | 0.135 | static |
| 5 | *distance from access points* | *0.224* | *static* |
| 6 | *distance from green spaces* | *0.223* | *static* |
| 7 | distance from water channels | 0.011 | static |
| 8 | *distance from roads* | *0.254* | *static* |
| 9 | distance from hydronodes | 0.065 | static |
| 10 | *distance from motorway junctions* | *0.161* | *static* |
| 11 | *distance from broadleaf forest* | *0.155* | *dynamic* |
| 12 | distance from conifer forest | 0.081 | dynamic |
| 13 | *distance from arable land* | *0.21* | *dynamic* |
| 14 | *distance from improved grassland* | *0.155* | *dynamic* |
| 15 | *distance from semi-natural grassland* | *0.161* | *dynamic* |
| 16 | distance from mountain, heath and bog | 0.013 | dynamic |
| 17 | *evidence likelihood of change from broadleaf forest to all other classes based on earlier land use transition* | *0.249* | *static* |
| 18 | *evidence likelihood of change from all other classes to broadleaf forest based on earlier land use transition* | *0.533* | *static* |
| 19 | *evidence likelihood of change from broadleaf forest to all other classes based on soil type* | *0.251* | *static* |
| 20 | *evidence likelihood of change from all other classes to broadleaf forest based on soil type* | *0.256* | *static* |

## 2.2.3. Transition sub-models (training and validation)

LCM enlists all shortlisted transitions between the two LULC maps, each represented by a transition sub-model. Explanatory variables which have Cramer's $V > 0.15$ were used to explain the observed transitions; the accuracy rates of the transition sub-models are given in electronic supplementary material, table S2. MLP procedure in LCM was then used to run the transition sub-models to empirically model future LULC. MLP used a backwards stepwise variable selection in which all variables are tested individually and in pairs for their impact on model accuracy and finally the likelihood of model overfitting is reduced by selecting an optimum number of variables to be included in the final model. When training and validating a transition sub-model, MLP makes use of sample pixels that have undergone a transition between the two time periods. By default, MLP takes 10 000 randomly selected pixels for running each transition sub-model. One half of these pixels is used to train the model, while the other half is used for model validation. At the end of each model training run, MLP generates accuracy results for each transition sub-model. The details of transition sub-models considered in this study and their respective accuracy are given in the electronic supplementary material, table S2. To project future changes, we generated a projected potential map—a map of the study area showing the potential of each pixel across the landscape to undergo each of the LULC transitions. The potential map was subsequently used to project LULC change to desired future date (2030).

## 2.2.4. Change demand modelling

In this step, we used the 'Change Demand Modelling' procedure in LCM to determine the amount of change that is likely to occur in selected LULC categories at some point in the future. By default, LCM

uses a Markov chain prediction process which calculates the amount of change based on historical observations and determines the area of land expected to undergo such transition in the future. At this stage, LCM generates a transition potential file in the form of a matrix which shows the probability of each LULC category to change to every other category. This potential file matrix generated by Markov chain showing probabilities of all transitions for the year 2030 is shown in table 2. LCM allows the user to manipulate these transition probabilities to create different modelling scenarios in the future. In this study, the following two scenarios for LULC transitions by 2030 were considered.

### 2.2.4.1. The business-as-usual scenario

This is the default scenario created in the Markov chain probability matrix. The business-as-usual (B-a-U) scenario represents a linear projection of current trends to 2030; the trends were identified and modelled based on changes observed between 2007 and 2015.

### 2.2.4.2. Ecosystem conservation scenario

By manipulating the transition probability matrix generated by the Change Demand Modelling panel, we created an ecosystem conservation (EC) scenario reflecting forestry and environmental policies currently in place:

(a) Conversion of conifer to broadleaf woodland in Wales would deliver a range of environmental benefits and is specifically required to deliver policies aimed at restoring PAWS. To accommodate these considerations in the EC scenario, we assumed that the probability of Conifer-to-Broadleaf Forest conversion would increase by 50% as compared to the current trend (B-a-U scenario).

(b) Conservation of ecologically important non-forest LULC categories such as Mountain, Heath and Bog is an integral part of Welsh forestry policy, for reasons of climate change in addition to biodiversity [53]. For this reason, we assumed that in 2015–2030, the probability of the LULC class 'Mountain, Heath and Bog' persisting itself will increase by 50% as compared to the current trend (B-a-U scenario).

(c) The Climate Change Strategy for Wales [20] inspired the Welsh Government to set a target of expanding existing Welsh woodland by 100 000 ha by 2030, although subsequently reduced to a minimum of 20 000 ha for this period [54]. Such expansion reflects wider aspirations for woodland expansion in the UK, e.g. in the 'UK 25 years environmental plan' [19]. We assumed, therefore, that the current trend of LULC classes such as Semi-Natural Grassland and Improved Grassland converting to Broadleaf Forest would continue, but Broadleaf Forest will not be converted to any other LULC category during 2015–2030.

### 2.2.5. Step 5: change projection

In the final step, we generated future LULC maps based on the transition probability matrices of the two scenarios considered in this study.

# 3. Results

## 3.1. LULC changes (2007–2015)

We computed LULC transitions between different LULC classes that have occurred in Wales, UK between 2007 and 2015 by applying the cross-tabulation module in LCM (figure 1a). The predictor variables considered in this study explained the LULC transitions between 2007 and 2015 well; the average accuracy of all transition sub-models considered in this study was 79% (electronic supplementary material, table S2). An accuracy rate of 75% or above is considered indicative of good model performance (Clark Labs, 2015). Gains and losses between different LULC types and net change calculation (figure 1a) show that 'Improved Grassland' and 'Broadleaf Forest' experienced the biggest expansion (148 521 and 39 875 ha, respectively). Moreover, an analysis of contributions to the net change of individual LULC types (electronic supplementary material, figure S1) suggests that the biggest contributor to 'Broadleaf Forest' during 2007–2015 is the 'Semi-Natural Grassland' category (19 589 ha). On the other hand, around 6458 ha of 'Broadleaf Forest' was lost to 'Coniferous Forest' during this time. Of the other major LULC changes in this period, around 80 778 ha of 'Semi-Natural Grassland' and 69 688 ha of 'Arable Land' were

**Table 2.** Markov chain transition probability matrix (business-as-usual scenario, ecosystem conservation scenario) showing probability of change to 2030 in Wales, UK.

| given | probability of changing to | | | | | |
|---|---|---|---|---|---|---|
| | broadleaf forest | conifer forest | arable land | improved grassland | semi-natural grassland | mountain, heath and bog |
| *business-as-usual scenario* | | | | | | |
| broadleaf forest | 0.570 | 0.142 | 0.019 | 0.191 | 0.062 | 0.013 |
| conifer forest | 0.188 | 0.754 | 0.003 | 0.029 | 0.020 | 0.004 |
| arable land | 0.052 | 0.009 | 0.098 | 0.752 | 0.079 | 0.007 |
| improved grassland | 0.040 | 0.008 | 0.057 | 0.784 | 0.103 | 0.006 |
| semi-natural grassland | 0.073 | 0.049 | 0.034 | 0.411 | 0.381 | 0.051 |
| mountain, heath and bog | 0.061 | 0.088 | 0.019 | 0.167 | 0.452 | 0.211 |
| *ecosystem conservation scenario* | | | | | | |
| broadleaf forest | 1 | 0 | 0 | 0 | 0 | 0 |
| conifer forest | 0.282 | 0.659 | 0.003 | 0.029 | 0.020 | 0.004 |
| arable land | 0.052 | 0.009 | 0.098 | 0.752 | 0.079 | 0.007 |
| improved grassland | 0.040 | 0.008 | 0.057 | 0.784 | 0.103 | 0.006 |
| semi-natural grassland | 0.073 | 0.049 | 0.034 | 0.411 | 0.381 | 0.051 |
| mountain, heath and bog | 0.061 | 0.088 | 0.019 | 0.167 | 0.347 | 0.316 |

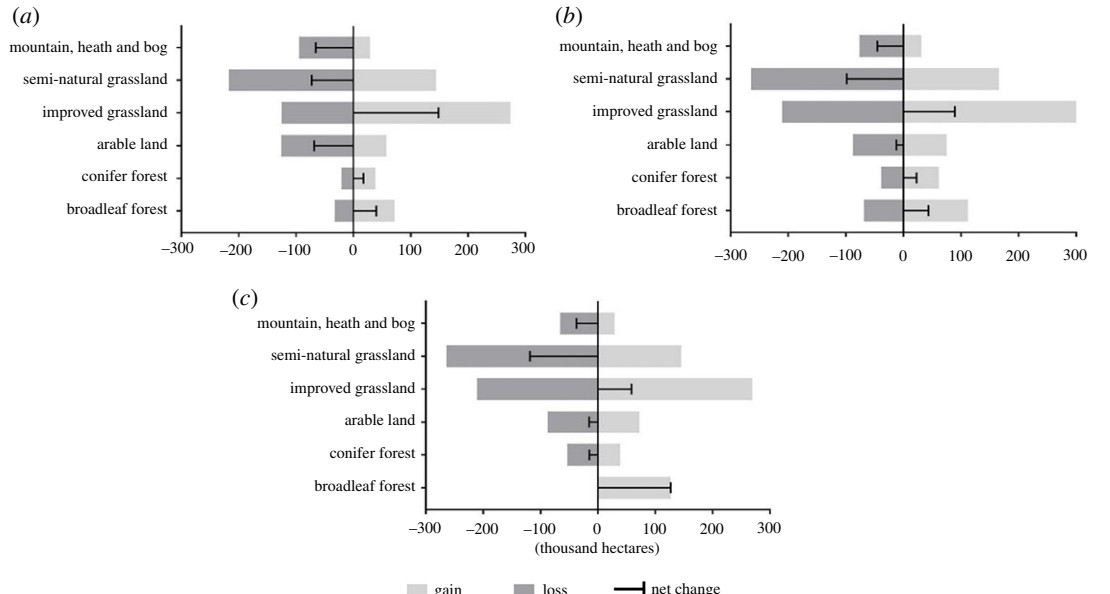

**Figure 1.** In Wales, UK, gains, losses and net changes between different LULC classes (hectares) during (*a*) 2007 – 2015, (*b*) 2015 – 2030 (B-a-U scenario) and (*c*) 2015 – 2030 (EC scenario).

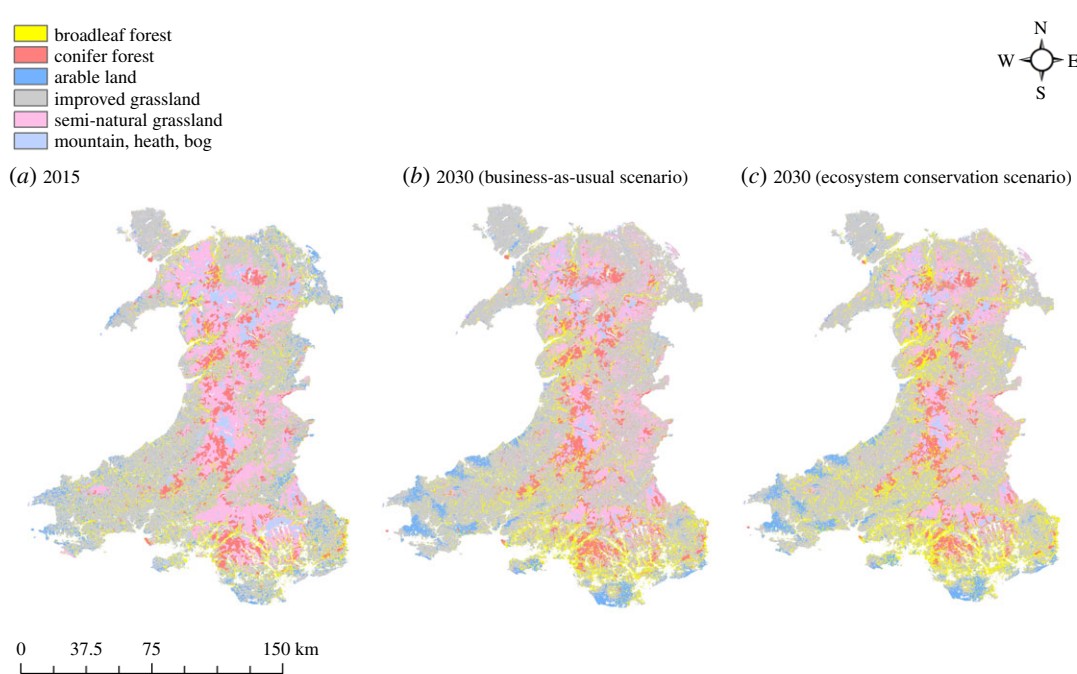

**Figure 2.** Current (*a*) and projected land use map of Wales, UK for the year 2030 under B-a-U (*b*) and EC (*c*) modelling scenarios.

converted to 'Improved Grassland'. A visual overview of historical and projected spatial distribution of LULC transitions in Wales is presented in electronic supplementary material, figure S2.

## 3.2. Land use change projections for 2030

### 3.2.1. The business-as-usual scenario

Based on LULC change observed over the period between 2007 and 2015, we generated a map showing projected LULC map of Wales for the year 2030 under the B-a-U scenario (figure 2) which is based on the projected potential for transition in Wales (electronic supplementary material, table S2 and figure S3). An analysis of the B-a-U scenario shows that the 'Broadleaf Forest' is likely to experience a net

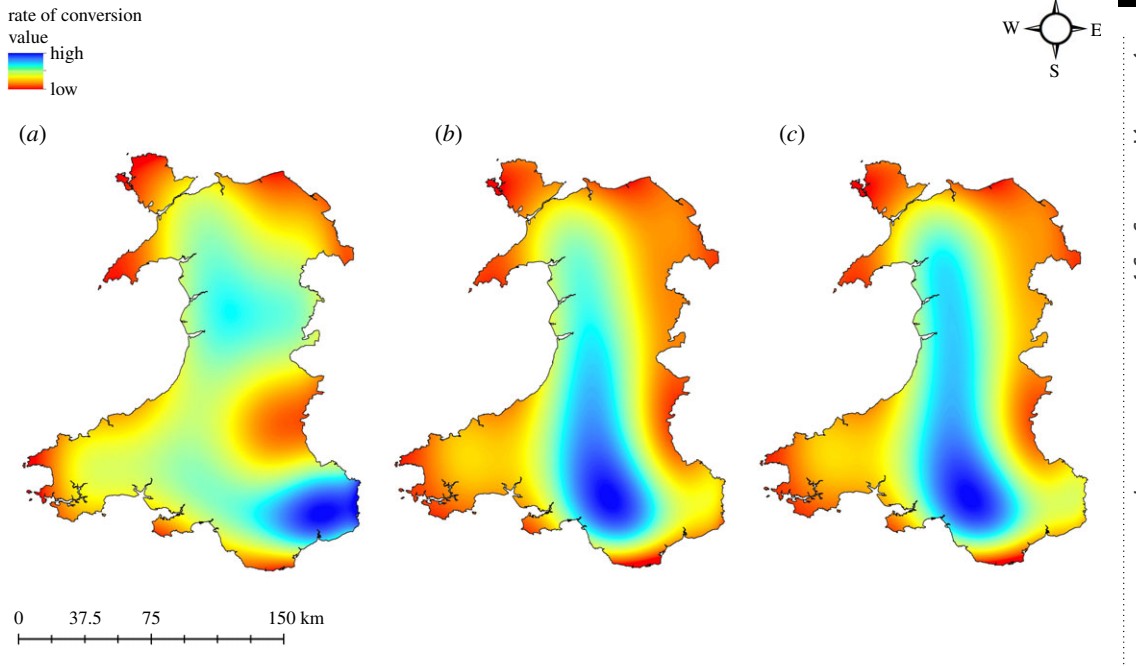

**Figure 3.** Spatial trend of conifer to broadleaf forest conversion in Wales, UK during 2007 – 2015 (*a*), 2015 – 2030 B-a-U scenario (*b*) and 2015 – 2030 EC scenario (*c*).

increase of 43 366 ha of area (figure 1*b*), most of which is likely to come at the expense of 'Semi-Natural Grassland' (21 321 ha), 'Improved Grassland' (8815 ha) and 'Coniferous Forest' (7480 ha).

### 3.2.2. The ecosystem conservation scenario

Projected transition potential map for the second scenario considered in this study is shown in electronic supplementary material, figure S3. An analysis of the LULC transition under this modelling scenario reveals that 'Broadleaf Forest' LULC category is expected to gain 127 129 ha of the area between 2015 and 2030 (figure 1*c*). Most of the expansion in the area of 'Broadleaf Forest' under this modelling scenario is expected to come at the cost of 'Coniferous Forest' (45 322 ha), 'Improved Grassland' (39 494 ha) and 'Semi-Natural Grassland' (31 318 ha).

## 3.3. State of forests in Wales in 2030

Compared to the baseline of 159 951 ha in 2015, the 'Broadleaf Forest' category is likely to expand to 203 317 and 287 080 ha under the B-a-U and EC modelling scenarios, respectively. Existing conifer woodlands are expected to experience a conversion to broadleaf of 19% and 28% under the B-a-U and EC scenarios, respectively. The total forest area in Wales (combining broadleaf and conifer forests) is expected to increase by 66 358 and 112 280 ha under the B-a-U and EC scenarios, as compared to the total area under forests in 2015. Historically (2007–2015), most of the conversion took place in the southeastern part of Wales. A slight modification of the geographical distribution of conversion is expected in both future scenarios modelled in this study: the contour map of conversion intensity indicates a westward and northward shift (figure 3). As an example, showing projected change of the area of the broadleaf forest against topographic detail, we show the detail of an area of Snowdonia National Park in Wales in figure 4.

## 4. Discussion

Forests covered nearly 15% of the total area of Wales in 2015. Most Welsh forests were privately owned (59%), the remainder was owned by the Welsh Government Woodland Estate [55]. The forestry sector makes a significant contribution to the Welsh economy. Recent data (2015–2016) indicate that this sector contributes a gross value added of £528.6 million, supports around 700 businesses and

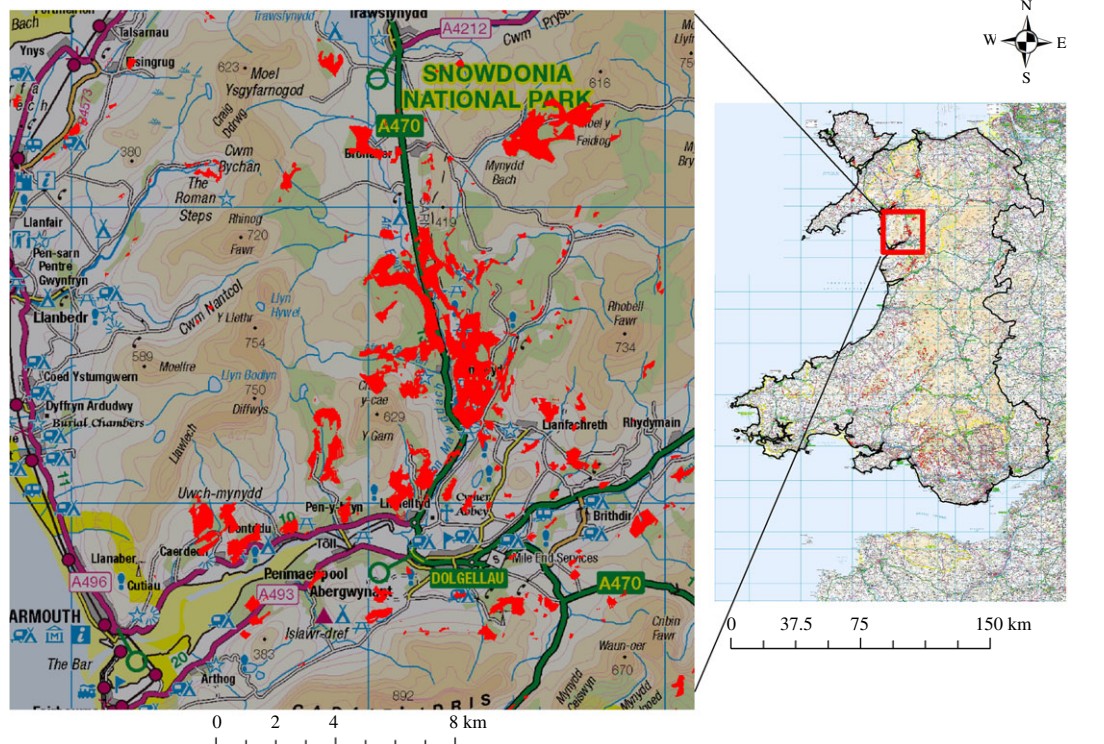

**Figure 4.** Projected conifer to broadleaf forest conversion in Wales during 2015 and 2030 under the EC modelling scenario. Broadleaf forest expansion in Snowdonia National Park is enlarged for detailed view (OS Crown copyright Edina Digimap).

provides over 10 000 jobs. Overall, the value of publicly owned forest in Wales in 2015 was estimated to be £642 million [55]. Besides their economic value, Welsh woodlands provide a range of ecosystem services by sequestering 1 419 000 tonnes of carbon dioxide equivalent per year and by playing a crucial role in soil and water management by reducing nutrient runoff, diffusing pollution, reducing flood risk and improving water quality [18]. Recent forest planting in Wales, however, has changed its focus from the twentieth century, the planting of broadleaf tree species increasing in comparison to conifers. Since 2001, the estimated area under conifer forests has decreased by 18 000 ha, while the estimated area under broadleaf species has increased by 35 000 ha [56].

The forested land area in Wales is very low for a European country, the average forest cover in Europe being 37% [18]. In order to increase the amount of land under forests, with the main motivation being ecosystem service provision, the Climate Change Strategy for Wales prompted the Welsh Government to set a target of increasing forest from 15% up to 19%; because of perceived the low levels of planting this target has been progressively reduced, and currently is the equivalent 20 000 ha by 2030 [54]. At the same time, the conversion of non-native conifers to native broadleaf species, primarily in the PAWS, is an aim of the UKFS [57]. In our B-a-U scenario, we found that if the current rate of change continues, the total forest area in Wales by 2030 is likely to increase from current 15 to 17%. However, in the EC scenario where the rates of afforestation and conversion to broadleaf are prioritized, future forest cover could reach the original target: 19% of the total Welsh area. This is considerably in excess of Welsh Government's revised target. This disparity is interesting and highlights the need for more informed projections of forest cover to help set policy targets. Historical conifer-to-broadleaf conversion rate projected to 2030 indicates that about 19% of existing conifers will undergo conversion, the EC scenario increases the rate by half to 28%. In standard forestry practice in Wales, where the average rotation of non-native conifers in Wales is about 60 years, the expectation is to harvest one-sixth of these woodlands every decade. The modelled B-a-U conversion rate of 19% is thus very close to the natural rotation harvest and replanting. Aiming for a higher conversion rate by 2030 implies shortening the rotation in some forests.

Although the major policy driver of conifer-to-broadleaf woodland conversion in Wales is PAWS, there are other factors that may motivate this type of conversion. For instance, the UKFS encourages diversifying forested areas in a way that a forest management unit should have at least 5% of native broadleaf

trees species and should not contain more than 75% of a single species [57]. The UKFS guidelines also advocate large-scale conversion in areas with the potential to enhance existing ancient semi-natural woodlands and on sites which are sufficiently large to overcome edge effects [57]. In a survey of private woodland managers in England, Scotland and Wales tasked with managing planted non-native conifer woodlands (but excluding PAWS), the managers were interviewed about their intentions regarding conversion to broadleaves and the reasons behind their plans. The results suggested that woodland managers are considering a conversion of anywhere between 5 and 95% of their woodlands from conifers to native broadleaf forest, even if not required to do so by environmental policy. Those willing to convert indicated biodiversity conservation, improved resilience and recreation as the chief factors behind their intention, while those unwilling to make large-scale conversions mentioned timber production, cost of conversion and—crucially—lack of guidance and advice [9].

In the recent past, considerable research has been carried out to improve understanding of costs and benefit of converting PAWS to the native broadleaf forest; however, no dedicated effort has been made on the conversion of non-PAWS, leaving a potential research gap [9]. Since the conversion needs to be gradual and well planned [9,58], our study presents a useful spatially explicit decision-making aid that pinpoints sites most suitable for conifer-to-broadleaf conversion across Wales. Projected maps of conifer-to-broadleaf conversion may also prove useful for forest managers who are hesitant of conversion by indicating site suitability on the basis of past experience. Concerns such as suitability of site and distance to already existing broadleaf seed source have been indicated by forest managers [9]; our projected LULC models address these concerns as we incorporate past LULC changes, soil factors, slope, water availability, road access and distance to the broadleaf forest as predictor variables in the models.

Analysis of the conifer-to-broadleaf forest conversion between 2007 and 2015 indicated that altitude is the most influential factor, altitude alone resulting in an accuracy of 80% of the 'conifer forest to broadleaf forest' sub-model. Most of the transitions from conifer plantation to broadleaf forests in Wales took place between altitudes of 100 and 250 m. Conversion was minimal close to sealevel and above 275 m. Most of the land between 0 and 100 m of altitude is dedicated to agriculture and grasslands, while forests are less likely to occur above 275 m in Wales as land use above this altitude predominantly belongs to the Mountain, Heath and Bog category. In addition to altitude, the distance from roads made a notable contribution in explaining the conifer-to-broadleaf conversion between 2007 and 2015. Most of the conifers converted to broadleaves were located in areas 200–300 m from a road. Habitat expansion or restoration activity is more likely to be within 1 km of the road network as easy and frequent access is generally possible in this range [52].

One of the motivations for designing the EC scenario in this study was conservation of the 'Mountain, Heath and Bog' habitat. Heaths and peat bogs include all inland and coastal, dry and wet heaths and mires [59]. These are some of Britain's most scarce habitats having a unique ecological value as these habitat types support a range of animals, insects and plants and provide food and shelter to migrating birds [60]. Moreover, heath and peat bogs—which can be thousands of years old—contain a wealth of historical data on climate, landscape and biodiversity. A study in 1995 had indicated that around 20% of upland heather moorland present in England and Wales in the mid-1940s was lost by 1990, high grazing pressure being one of the chief reasons for this loss [60].

A recent study modelled species richness in heath and peat bog habitat in the UK and warned of major declines in this habitat by 2030 [59]. Our data show that 70% of Welsh heather moorland present in 1990s is 'at risk of change' in future. During 2007–2015, 40% of 'Mountain, Heath and Bog' was lost, three-quarters of the loss due to the conversion to 'Semi-Natural Grassland' category. Under the B-a-U scenario, 47% of the 2015 area under this LULC class is likely to be lost by 2030. Under the EC scenario—where we increased the probability of 'Mountain, Heath and Bog' to persist by 50%—we still saw a 38% loss. This indication of serious threats to 'Mountain, Heath and Bog' habitat suggests that major interventions are required to conserve this highly valuable and scarce habitat.

# 5. Applications of future land use and land cover maps for ecosystem services and biodiversity analysis in Wales

In addition to the utility as a decision support tool for broadleaf forest expansion in Wales, the output LULC maps of this study can be used for a range of habitat, ecosystem services and biodiversity analysis, some of which are discussed in the following.

## 5.1. Modelling the abundance of crop pollinators

It is estimated that 20% of agricultural crops in the UK depend on pollinators; the economic value of pollinators to UK agriculture is estimated to be £690 million per year [61]. The National Ecosystem Assessment showed that the abundance of wild pollinators in the UK has declined in the last 30 years and this trend is likely to continue [62]. In Wales, the declining abundance of pollinators is an increasingly important issue and several action plans have been proposed in the recent past to conserve pollinators [61,63]. The future LULC maps of Wales can be used to evaluate the likely effects of contrasting LULC scenarios on the future abundance of wild bees as pollinators for agricultural crops. The abundance of wild bees largely depends on factors like nesting site availability and flight range, and models based on current and future LULC maps can be used to estimate the number of wild bees visiting agricultural sites and identifying areas that are likely to gain the most benefits from wild pollination [47].

## 5.2. Predicting carbon storage and sequestration

The Welsh Government has a commitment to reduce annual carbon equivalent emissions by 3% per annum [64,65]. In this regard, monitoring the rates of carbon emissions and sequestration is a challenge. Carbon sequestration models are being used to estimate future carbon stocks and rates of sequestration [66]. The future LULC maps of Wales can be used to estimate future amounts of carbon stored in the landscape and sequestered over time. Current and future LULC maps along with rates of wood harvest, degradation rates of harvested products and carbon stocks in the current LULC classes can be used to calculate carbon storage and sequestration. Additionally, these maps can also be used to calculate the market value of sequestered carbon in current and future land cover of Wales [47].

## 5.3. Future landscape change process analysis

As stated in the State of Nature (2016) report for Wales, major changes in ecosystem services and abundance of species have occurred due to degradation and fragmentation of habitat in Wales [67]. Current and projected future LULC maps of Wales can be used to measure the nature of the change underway within each land use class under different policy scenarios of broadleaf forest expansion in Wales. In spatial analysis environments such as TerrSet, it is done by using a decision tree method that compares the land cover patches in each LULC category between the two time periods and calculates the changes in the perimeter and area of the corresponding patches [47]. The output map helps visualizing where in future a given LULC category is likely to experience persistence, fragmentation or aggregation of patches. In the context of Wales where there is a major drive to conserve and repair rare natural habitat such as bogs, this could be an important analysis as it would allow one to visualize the nature of landscape change that might occur under different future land use change scenarios.

## 5.4. Invasive species distribution modelling in Wales

Invasive species are major drivers of ecosystem degradation in Wales costing the Welsh economy approximately £7 million per annum [68]. Predicting the future distribution of invasive species is key to effective invasive species management and planning [69]. Future distribution of invasive species is often governed by land cover type. Evidence suggests that in Wales the spread of invasive species such as *Rhododendron ponticum* is more sensitive to the land cover type than any other biophysical or climatic factor [69]. The projected future maps of our study can be used to run species distribution models of different invasive as well as any other species of interest. Contrasting the future scenarios may help understand how the different policy approaches are likely to affect the course of invasive species distribution in Wales.

# 6. Limitations of the study

Explanatory variables used in land use change modelling studies are generally divided into three categories: biophysical, proximate and socio-economic variables [42]. In this study, we did not include socio-economic variables owing to the coarse resolution of available datasets. Instead, we preserved the fine spatial resolution of biophysical and proximate variables which are likely to be stronger determinants of land use change and reasonable proxies for the socio-economic variables. However, it is advisable to bear in

mind that the resulting future projection does not directly represent the socio-economic landscape. Availability of fine-scale socio-economic and climatic variables may improve the modelling outputs in the future. Moreover, we used LULC maps generated by the Centre for Ecology & Hydrology, UK [41] which, to the best of our knowledge, are currently the most accurate, verified and finest-resolution LULC temporally repeated maps covering the UK. The fact that we used only two time points is a limitation of our study; we did not have a map of LULC after 2015 that could have been used for verification of the future projection. Although we adopted published protocols for future LULC projections [70,71], we suggest that the use of three or more historical LULC time points be considered for verification of projections. As we gather more archived LULC data, this approach should become the norm. Finally, a key limitation of this type of analysis is its 'blindness' to major shifts of socio-economic landscape and hence its inability to factor these into projections. A case in point is Brexit, where a set of self-imposed trade sanctions threatens a severe adjustment of existing drivers of land use.

# 7. Conclusion

This study reveals the changes in LULC in Wales from 2007 to 2015 by using a combined approach of GIS and land change prediction models. An integrated MLP–MCA method was applied to improve understanding of the scale and location of probable LULC changes under linear projection (B-a-U) and a policy-based future scenario (EC) up to 2030. Broadleaf forest expansion is likely to reach the targets set by the Welsh Government under the EC scenario. The study shows the potential of LULC predictions to test alternative policy aims and to generate evidence at a scale useful to local decision makers. This type of tool can contribute to sustainable development by providing an evidence-based spatial framework to support restoration and conservation of ecologically important habitats in Wales. Since land use and land cover change is a highly complex phenomenon affected by a range of ecological, political and socio-economic factors, we contend that models incorporating the widest range of factors be used to test future LULC scenarios.

Data accessibility. The land cover maps of Wales used in this study were downloaded from Edina Digimap Environment Products https://digimap.edina.ac.uk/environment. The explanatory variables used in the study were downloaded from the Ordnance Survey https://digimap.edina.ac.uk/os. The software used for land use change modelling was acquired from Clark Labs https://clarklabs.org/terrset/.

Authors' contributions. S.A.M. and G.G. carried out the spatial analysis and participated in land use change modelling. M.L. and J.L. designed and discussed the contrasting future scenarios for the land use change modelling in Wales in the light of Welsh forest policy. S.A.M., G.G., M.L. and J.L. conceived the study, designed the study, coordinated the study and helped draft the manuscript. All authors gave final approval for publication.

Competing interests. We declare we have no competing interests.

Funding. This PhD study was funded by the Commonwealth Scholarship Commission (2016-86).

Acknowledgements. We thank Dr Kotaro Iizuka (University of Tokyo) for his advice on land use change modelling.

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
