## [Reviewer comments · Royal Society Open Science]

Review History

RSOS-190026.R0 (Original submission)

Review form: Reviewer 1

Is the manuscript scientifically sound in its present form?

Yes

Are the interpretations and conclusions justified by the results?

Yes

Is the language acceptable?

Yes

Is it clear how to access all supporting data?

Yes

Do you have any ethical concerns with this paper?

No

Have you any concerns about statistical analyses in this paper?

No

Recommendation?

Accept with minor revision (please list in comments)

Comments to the Author(s)

The authors used a land cover change model to simulate future broadleaf expansion in Wales under two scenarios. I found the paper very interesting to read and very well done. I have only some minor suggestions before recommending it to publication. In detail:

- line 89: I would argue against using the word "predict" when it comes to modeling future land cover change. In reality these models can "project", but not really "predict" as they are blind to anticipate several socio-economic dynamics that surely impact future land cover change (e.g. predicting the impact of Brexit). Also, it would be important to add a sentence to the introduction on how these models assume that the relationships found between the explanatory variables and the observed change remains stable over time, which is also not the case in real life (socio-economic dynamics tend to vary over time, e.g. dietary changes vary over time leading to different impacts on land use dynamics; focus on restoration may increase due to policy changes).
- line 105: I would also argue that these models are not necessarily providing a "high level of accuracy". It depends on how these are (usually poorly) validated. If you compare projected with observed and considered all the landscape, surely the accuracy is high because 90% of the landscape doesn't change. If you compare observed and predicted CHANGE, then their accuracy is quite low actually (see, for example, Rosa et al. 2014 Global Change Biology).
- I was surprised that the authors did not attempt to test the correlation between the explanatory variables. Surely, Distance to Access Points and Distance from Roads are correlated variables. Is this not the case?
- line 222: I would recommend avoiding qualitative statements such as "fairly well", because to me, a 50% accuracy is not a good accuracy. Also, given that you know how well these variables explain the different transitions, why not just focus on the good ones to project the future? Or better yet, why not use the quality of the model (% accuracy) to weigh the projections? It seems a bit odd to me that then the authors mix the results of really good and bad models in the same map...
- It was also not clear to me how the authors converted the transition probabilities to an actual change in area (Figure 2). It would be important to clarify that under section 2.2.5. Was the highest transition probability in each pixel chosen to be the one projected to occur?
- line 295: where does the 79% accuracy come from? Is it the average of all models? If so, please add the standard error as well
- in Figure 1b, it is not clear to me why there is such a big difference between the BAU scenario and the historical changes in Improved Grassland... shouldn't it be relatively similar? Since BAU is an extrapolation of the historical trends.
- It's almost impossible to visualize anything in Figure 3... I would consider removing this figure as it doesn't seem to add much to the main messages of the manuscript. Or replace it by another type of visualization highlighting the main message, i.e. the areas where you historically had broadleaf expansion and not necessarily the same as those projected to occur in the future. Or combine existing figure with Figure 4, by placing the projected change (in black) on top off the potential for change.
- Figure 4 - is this displaying the rate of conversion (more area) or the potential for conversion (higher probability). It is unclear to me at the moment.
- I know it is a bit the "elephant in the room", but perhaps it would make sense to at least mention Brexit in the Discussion of the paper, and how its uncertainty might impact the British landscape. Further, it would highlight the useful characteristics of scenario modeling and scenario analysis for dealing with uncertainties such as those associated with Brexit.

Apart from the comments above, I am happy with the manuscript. I think it is very well written, the analysis is statistically sound, and the results and interpretation are well explained.

Review form: Reviewer 2

Is the manuscript scientifically sound in its present form?

No

Are the interpretations and conclusions justified by the results?

No

Is the language acceptable?

Yes

Is it clear how to access all supporting data?

Yes

Do you have any ethical concerns with this paper?

No

Have you any concerns about statistical analyses in this paper?

No

Recommendation?

Accept with minor revision (please list in comments)

Comments to the Author(s)

Overview:

The submitted manuscript describes an alternative futures scenario modeling study designed to understand how broadleaf forests and other natural ecosystems in Wales will change under different policy scenarios. The authors evaluate two scenarios, "business as usual" and a conservation scenario. The results of the modeling show that Broadleaf forests in Wales are likely to increase at a greater rate under the conservation scenario due to increased Conifer-to-Broadleaf conversion and reduced levels of Broadleaf loss. The results of the modeling yield a range of insights about the expected future distribution of future forest and grassland ecosystems, including that the current rate of forest expansion will meet current policy goals. The results of this study also provide a dataset that can be used to better understand expected future changes in related ecosystem services and biodiversity. The authors also highlight the seriousness of loss with respect to bog ecosystems and that new policy-based effort is needed urgently to address this situation. Notwithstanding the potential concerns discussed below, the future scenarios are well constructed and empirically rigorous. In particular, the variable selection process is robust and the accuracy assessment scores and process are sound. Also, the academic writing is strong and the background information and description of the modeling processes undertaken are well described.

Concerns/Suggestions:

My main concern is that the conservation scenario assumes that Broadleaf conversion will stop, but it is not stated how this would take place in terms of existing or expected policies. Specifically, the authors note there is a public policy goal to increase Broadleaf forest cover and a

general aspiration in support of this goal, but nothing is said about a parallel goal or aspiration to decrease Broadleaf forest loss. This model assumption can theoretically be acceptable, but adding to my confusion is how the authors expect this outcome to occur because related policy proposals or findings are not addressed in the scenario assumptions of the conservation model or in the discussion, where potential Broadleaf expansion policies are discussed, such as decreased harvesting rotations to accelerate the conversion of Conifer forests to Broadleaf forests.

All told, given that the rate of Broadleaf loss was significant 2007-2015, more attention should be given to (a) justifying how no new Broadleaf loss would occur through 2030 from a policy perspective and/or (b) offering recommendations based on this research or other research that indicates how ending the historic drivers of Broadleaf conversion can be achieved. In the absence of discussing either of these elements, the proposed conservation future scenario appears unlikely to be achievable and thus the model's results offer less value and accuracy than would be true otherwise.

The second item of concern is that the description of the assumption for the conservation transition probability for 'Mountain, Heath and Bog' appears to be cut off (Page 9 line 254). This means the adjustments made to the related transition probabilities in Table 2 cannot be fully understood or verified.

The last item of note is that some LULC categories and transitions were excluded because they were "not relevant to our objectives (e.g., urban areas)." I presume that urbanization was a factor in driving historic Broadleaf forest loss. Either way, this paper could be improved in supporting its policy recommendations by better describing the causes of ecosystem conversion historically and by offering more discussion about how these factors could/would be mitigated in the conservation scenario, especially given the conservation scenario assumption of no net Broadleaf forest loss. Also, if there was no significant historic loss of ecosystems due to "Built Areas," I suggest noting this in the paper.

Decision letter (RSOS-190026.R0)

08-Apr-2019

Dear Mr Manzoor

On behalf of the Editors, I am pleased to inform you that your Manuscript RSOS-190026 entitled "Scenario-led modelling of broadleaf forest expansion in Wales" has been accepted for publication in Royal Society Open Science subject to minor revision in accordance with the referee suggestions. Please find the referees' comments at the end of this email.

The reviewers and handling editors have recommended publication, but also suggest some minor revisions to your manuscript. Therefore, I invite you to respond to the comments and revise your manuscript.

- Ethics statement

- Data accessibility

If you wish to submit your supporting data or code to Dryad (<http://datadryad.org/>), or modify your current submission to dryad, please use the following link:
<http://datadryad.org/submit?journalID=RSOS&manu=RSOS-190026>

- Competing interests

- Authors' contributions

- Acknowledgements

- Funding statement

Because the schedule for publication is very tight, it is a condition of publication that you submit the revised version of your manuscript before 17-Apr-2019. Please note that the revision deadline

will expire at 00.00am on this date. If you do not think you will be able to meet this date please let me know immediately.

If your manuscript is newly submitted and subsequently accepted for publication, you will be asked to pay the article processing charge, unless you request a waiver and this is approved by

Royal Society Publishing. You can find out more about the charges at <http://rsos.royalsocietypublishing.org/page/charges>. Should you have any queries, please contact openscience@royalsociety.org.

on behalf of Dr Bethan Davies (Associate Editor) and Professor Jon Blundy (Subject Editor)
openscience@royalsociety.org

Reviewer comments to Author:
 Reviewer: 1

Comments to the Author(s)

The authors used a land cover change model to simulate future broadleaf expansion in Wales under two scenarios. I found the paper very interesting to read and very well done. I have only some minor suggestions before recommending it to publication. In detail:

- line 89: I would argue against using the word "predict" when it comes to modeling future land cover change. In reality these models can "project", but not really "predict" as they are blind to anticipate several socio-economic dynamics that surely impact future land cover change (e.g. predicting the impact of Brexit). Also, it would be important to add a sentence to the introduction on how these models assume that the relationships found between the explanatory variables and the observed change remains stable over time, which is also not the case in real life (socio-economic dynamics tend to vary over time, e.g. dietary changes vary over time leading to different impacts on land use dynamics; focus on restoration may increase due to policy changes).
- line 105: I would also argue that these models are not necessarily providing a "high level of accuracy". It depends on how these are (usually poorly) validated. If you compare projected with observed and considered all the landscape, surely the accuracy is high because 90% of the landscape doesn't change. If you compare observed and predicted CHANGE, then their accuracy is quite low actually (see, for example, Rosa et al. 2014 Global Change Biology).
- I was surprised that the authors did not attempt to test the correlation between the explanatory variables. Surely, Distance to Access Points and Distance from Roads are correlated variables. Is this not the case?
- line 222: I would recommend avoiding qualitative statements such as "fairly well", because to me, a 50% accuracy is not a good accuracy. Also, given that you know how well these variables explain the different transitions, why not just focus on the good ones to project the future? Or better yet, why not use the quality of the model (% accuracy) to weigh the projections? It seems a bit odd to me that then the authors mix the results of really good and bad models in the same map...
- It was also not clear to me how the authors converted the transition probabilities to an actual change in area (Figure 2). It would be important to clarify that under section 2.2.5. Was the highest transition probability in each pixel chosen to be the one projected to occur?
- line 295: where does the 79% accuracy come from? Is it the average of all models? If so, please add the standard error as well

- in Figure 1b, it is not clear to me why there is such a big difference between the BAU scenario and the historical changes in Improved Grassland... shouldn't it be relatively similar? Since BAU is an extrapolation of the historical trends.

- It's almost impossible to visualize anything in Figure 3... I would consider removing this figure as it doesn't seem to add much to the main messages of the manuscript. Or replace it by another type of visualization highlighting the main message, i.e. the areas where you historically had broadleaf expansion and not necessarily the same as those projected to occur in the future. Or combine existing figure with Figure 4, by placing the projected change (in black) on top off the potential for change.

- Figure 4 - is this displaying the rate of conversion (more area) or the potential for conversion (higher probability). It is unclear to me at the moment.

- I know it is a bit the "elephant in the room", but perhaps it would make sense to at least mention Brexit in the Discussion of the paper, and how its uncertainty might impact the British landscape. Further, it would highlight the useful characteristics of scenario modeling and scenario analysis for dealing with uncertainties such as those associated with Brexit.

Apart from the comments above, I am happy with the manuscript. I think it is very well written, the analysis is statistically sound, and the results and interpretation are well explained.

Reviewer: 2

Comments to the Author(s)

Overview:

The submitted manuscript describes an alternative futures scenario modeling study designed to understand how broadleaf forests and other natural ecosystems in Wales will change under different policy scenarios. The authors evaluate two scenarios, "business as usual" and a conservation scenario. The results of the modeling show that Broadleaf forests in Wales are likely to increase at a greater rate under the conservation scenario due to increased Conifer-to-Broadleaf conversion and reduced levels of Broadleaf loss. The results of the modeling yield a range of insights about the expected future distribution of future forest and grassland ecosystems, including that the current rate of forest expansion will meet current policy goals. The results of this study also provide a dataset that can be used to better understand expected future changes in related ecosystem services and biodiversity. The authors also highlight the seriousness of loss with respect to bog ecosystems and that new policy-based effort is needed urgently to address this situation. Notwithstanding the potential concerns discussed below, the future scenarios are well constructed and empirically rigorous. In particular, the variable selection process is robust and the accuracy assessment scores and process are sound. Also, the academic writing is strong and the background information and description of the modeling processes undertaken are well described.

Concerns/Suggestions:

My main concern is that the conservation scenario assumes that Broadleaf conversion will stop, but it is not stated how this would take place in terms of existing or expected policies. Specifically, the authors note there is a public policy goal to increase Broadleaf forest cover and a general aspiration in support of this goal, but nothing is said about a parallel goal or aspiration to decrease Broadleaf forest loss. This model assumption can theoretically be acceptable, but adding to my confusion is how the authors expect this outcome to occur because related policy proposals or findings are not addressed in the scenario assumptions of the conservation model or in the discussion, where potential Broadleaf expansion policies are discussed, such as decreased harvesting rotations to accelerate the conversion of Conifer forests to Broadleaf forests.

All told, given that the rate of Broadleaf loss was significant 2007-2015, more attention should be given to (a) justifying how no new Broadleaf loss would occur through 2030 from a policy

perspective and/or (b) offering recommendations based on this research or other research that indicates how ending the historic drivers of Broadleaf conversion can be achieved. In the absence of discussing either of these elements, the proposed conservation future scenario appears unlikely to be achievable and thus the model's results offer less value and accuracy than would be true otherwise.

The second item of concern is that the description of the assumption for the conservation transition probability for 'Mountain, Heath and Bog' appears to be cut off (Page 9 line 254). This means the adjustments made to the related transition probabilities in Table 2 cannot be fully understood or verified.

The last item of note is that some LULC categories and transitions were excluded because they were "not relevant to our objectives (e.g., urban areas)." I presume that urbanization was a factor in driving historic Broadleaf forest loss. Either way, this paper could be improved in supporting its policy recommendations by better describing the causes of ecosystem conversion historically and by offering more discussion about how these factors could/would be mitigated in the conservation scenario, especially given the conservation scenario assumption of no net Broadleaf forest loss. Also, if there was no significant historic loss of ecosystems due to "Built Areas," I suggest noting this in the paper.

Author's Response to Decision Letter for (RSOS-190026.R0)

See Appendix A.

Decision letter (RSOS-190026.R1)

12-Apr-2019

Dear Mr Manzoor,

I am pleased to inform you that your manuscript entitled "Scenario-led modelling of broadleaf forest expansion in Wales" is now accepted for publication in Royal Society Open Science.

on behalf of Dr Bethan Davies (Associate Editor) and Jon Blundy (Subject Editor)
openscience@royalsociety.org

Follow Royal Society Publishing on Twitter: [@RSocPublishing](https://twitter.com/RSocPublishing)
Follow Royal Society Publishing on Facebook:
<https://www.facebook.com/RoyalSocietyPublishing.FanPage/>
Read Royal Society Publishing's blog: <https://blogs.royalsociety.org/publishing/>

Appendix A

Reviewer 1

Comment 1: line 89: I would argue against using the word "predict" when it comes to modelling future land cover change. In reality these models can "project", but not really "predict" as they are blind to anticipate several socio-economic dynamics that surely impact future land cover change (e.g. predicting the impact of Brexit). Also, it would be important to add a sentence to the introduction on how these models assume that the relationships found between the explanatory variables and the observed change remains stable over time, which is also not the case in real life (socio-economic dynamics tend to vary over time, e.g. dietary changes vary over time leading to different impacts on land use dynamics; focus on restoration may increase due to policy changes).

Response: As rightly pointed out, the difference in *predict* and *project* is important to understand. Changes made in line 90 and in other parts of the manuscript where *project* is a more suitable term in place of *predict*.

The point about the dynamic nature of socio-economic drivers is a very good one and we have now highlighted this in the paper (lines 110-112). Moreover, as explained in lines 198-202, variables can be both static and dynamic. Values of a dynamic variables change at each time step of the model run and thus need to be recalculated (e.g. the distance from broadleaf forest as these forests expand). By contrast, static variables remain constant over time (e.g. altitude, slope, soil type). In Table 1, all variables are labelled as either static or dynamic.

Comment 2: line 105: I would also argue that these models are not necessarily providing a "high level of accuracy". It depends on how these are (usually poorly) validated. If you compare projected with observed and considered all the landscape, surely the accuracy is high because 90% of the landscape doesn't change. If you compare observed and predicted CHANGE, then their accuracy is quite low actually (see, for example, Rosa et al. 2014 Global Change Biology.)

Response: Changes made in line 106-107, as suggested.

Comment 3: I was surprised that the authors did not attempt to test the correlation between the explanatory variables. Surely, Distance to Access Points and Distance from Roads are correlated variables. Is this not the case?

Response: In this study, we used Multilayer-Perceptron (MLP) algorithm, and following the instructions in the user manual of the Terrset software, we included all possible variables likely to explain the past transitions and then selected the variables in two steps: In the first step, we dropped variables of low explanatory power based on Cremer V value and in the second step the MLP used a backwards stepwise variable selection to use the minimum number of least correlated variables to be included in the final model (added to line 231-233 now). The second step thus corresponds to testing the correlation between variables to reduce model overfitting. However, could have been interesting to manually test for correlation and compare the two methods to see if a different selection of variables would be suggested.

Comment 4: line 222: I would recommend avoiding qualitative statements such as "fairly well", because to me, a 50% accuracy is not a good accuracy. Also, given that you know how well these variables explain the different transitions, why not just focus on the good ones to project the future? Or better yet, why not use the quality of the model (% accuracy) to weigh the projections? It seems a bit odd to me that then the authors mix the results of really good and bad models in the same map...

Response: Changes made as suggested (lines 227-228). We included all possible variables likely to explain the land use change. The Multilayer-Perceptron algorithm used a stepwise selection of variables for each transition sub-model, retaining the most powerful variables. Therefore, instead

of selecting variables for individual land use changes, we automated this process by letting the algorithm choose the best set of variables to explain each change.

There are 2 sub-models which failed to accurately predict some transitions (for example, broadleaf to improved and semi-natural grasslands were predicted with an accuracy of 50%). Luckily, these sub-models were of low importance since only a small land area changed from broadleaf forest to these two classes (and this is actually the reason why model couldn't predict them accurately since only a few pixels were available for training and testing). But despite being inaccurate, we had to include these sub-models in the final ensemble, or the model would consider no transition between these classes. But since our main focus was on the transitions to and from broadleaf and conifer forest, we included these sub-models in the final model.

Comment 5: It was also not clear to me how the authors converted the transition probabilities to an actual change in area (Figure 2). It would be important to clarify that under section 2.2.5. Was the highest transition probability in each pixel chosen to be the one projected to occur?

Response: The transition probability matrix is calculated from the transition sub models (table 1) and this matrix is then translated into transition potential maps (supplementary data S2, Fig S3.). The transition potential map is used to allocate land class to each pixel.

We used the Markov Chain algorithm to determine the amount of change that will occur until 2030. The Markov Chain determines the amount of change using the earlier and later landcover maps along with the date specified. The procedure determines how much land would be expected to transition from the later date to the prediction date based on a projection of the transition potentials into the future and creates a transition probabilities file. The transition probabilities file is a matrix that records the probability that each landcover category will change to every other category.

Comment 6: line 295: where does the 79% accuracy come from? Is it the average of all models? If so, please add the standard error as well

Response: 79% is the average of the accuracies of the sub-models of the transitions among all the classes considered in this study (supplementary data S1, Table S2), we explain this point now in lines 306-307.

Comment 7: In Figure 1b, it is not clear to me why there is such a big difference between the BAU scenario and the historical changes in Improved Grassland... shouldn't it be relatively similar? Since BAU is an extrapolation of the historical trends.

Response: This is an interesting observation. Clearly, the area gained/lost by Improved Grasslands is much more between 2015-2030 (B.A.U) as compared to the past (2007-2015). In Wales, Improved grassland and semi-natural grassland are the largest land use classes occupying nearly 70% of the Welsh landscape. Between 2007-2015, the largest contribution to the expansion of improved grassland came from the semi-natural grassland. If this trend continues (as assumed in the B.A.U scenario) the change in the area would have the same direction (net change positive) but the amount of change would be larger as compared to past since the period of projection is double the time of the past analysis (the past analysis shows area gained/lost in 8 years while the future projections show area likely to be gained/lost in next 15 years). Therefore, we would expect the extrapolation to show a net positive gain in the area (as it shows) but a larger number of hectares in terms of transition occurring.

Comment 8: It's almost impossible to visualize anything in Figure 3... I would consider removing this figure as it doesn't seem to add much to the main messages of the manuscript. Or replace it by

another type of visualization highlighting the main message, i.e. the areas where you historically had broadleaf expansion and not necessarily the same as those projected to occur in the future. Or combine existing figure with Figure 4, by placing the projected change (in black) on top off the potential for change.

Response: As suggested, figure 3 has been removed from the manuscript.

Comment 9: Figure 4 - is this displaying the rate of conversion (more area) or the potential for conversion (higher probability). It is unclear to me at the moment.

Response: Figure 4 shows the spatial trend of conifer to broadleaf forest conversion in Wales. The intention of this figure is to provide a means of generalizing about the pattern of change. The numeric values (0-1) do not have any special significance. The surface is created by coding areas of change with 1 and areas of no change with 0 and treating them as if they were quantitative values. In other words, this figure is a qualitative representation of the quantitative change (number of pixels which underwent change).

Comment 10: I know it is a bit the "elephant in the room", but perhaps it would make sense to at least mention Brexit in the Discussion of the paper, and how its uncertainty might impact the British landscape. Further, it would highlight the useful characteristics of scenario modelling and scenario analysis for dealing with uncertainties such as those associated with Brexit.

Response: One of the scenarios we originally considered (and may still attempt to run, even though the likelihood of any Brexit at all is receding fact) was projecting land use in Wales in the event of a near-complete collapse of sheep farming. We have added a short description of this to the discussion, it is an important point (lines 529-532).

Reviewer 2

Comment 1: My main concern is that the conservation scenario assumes that Broadleaf conversion will stop, but it is not stated how this would take place in terms of existing or expected policies. Specifically, the authors note there is a public policy goal to increase Broadleaf forest cover and a general aspiration in support of this goal, but nothing is said about a parallel goal or aspiration to decrease Broadleaf forest loss. This model assumption can theoretically be acceptable, but adding to my confusion is how the authors expect this outcome to occur because related policy proposals or findings are not addressed in the scenario assumptions of the conservation model or in the discussion, where potential Broadleaf expansion policies are discussed, such as decreased harvesting rotations to accelerate the conversion of Conifer forests to Broadleaf forests.

Response: The E.C (ecosystem conservation scenario) assumes an extreme case where other land use categories will continue to convert to Broadleaf forest, but the existing broadleaf forest will remain intact (no more deforestation). Of course, such this scenario is practically nearly impossible, but it meant to give the theoretically maximum possible conservation/increase of broadleaf forest in future. We have highlighted this point in the paper now as it is an important one. To our knowledge, stated policy is to expand broadleaf woodland. We assume the corollary, that there is an intention not to allow further losses to existing broadleaf woodland.

Comment 2: All told, given that the rate of Broadleaf loss was significant 2007-2015, more attention

should be given to (a) justifying how no new Broadleaf loss would occur through 2030 from a policy perspective and/or (b) offering recommendations based on this research or other research that indicates how ending the historic drivers of Broadleaf conversion can be achieved. In the absence of discussing either of these elements, the proposed conservation future scenario appears unlikely to be achievable and thus the model's results offer less value and accuracy than would be true otherwise.

Response: It is emphasized in forest policy statements that without significantly reducing the rates of deforestation, there is a danger of net deforestation occurring, given that new woodland creation is continuing only at a low rate [1]. Moreover, to abide by the international commitments to prevent deforestation, the Forestry Acts (since 2010) declare it illegal to remove more than 5 cubic meters of growing trees per quarter in Great Britain, without approval [1]. In addition to this, different organizations working on developing a network of healthy and resilient ecosystems in the UK (i.e. The Scottish Wildlife Trust) focus on expanding the existing communities of native species by restoring the degraded habitats and protecting the existing habitat from further loss [1]. Therefore, the current policies revolve around the notion of decreasing the deforestation and the assumption of "no broadleaf deforestation" in the E.C. scenario – though only theoretical and perhaps not achievable by 2030 – makes a sense just to visualize the possible outcomes of the maximum broadleaf forest protection in Wales.

1. Atkinson S, Townsend M. 2011 The State of the UK's Forests, Woods and Trees.

Comment 3: The second item of concern is that the description of the assumption for the conservation transition probability for 'Mountain, Heath and Bog' appears to be cut off (Page 9 line 254). This means the adjustments made to the related transition probabilities in Table 2 cannot be fully understood or verified.

Response: Thank you for pointing out this extremely important mistake. It was a typing error. We assumed that in 2015-2030, the probability of the LULC class, 'Mountain, Heath and Bog' persisting itself will increase by 50%. The changes have been made in line 273-274.

Comment 4: The last item of note is that some LULC categories and transitions were excluded because they were "not relevant to our objectives (e.g., urban areas)." I presume that urbanization was a factor in driving historic Broadleaf forest loss. Either way, this paper could be improved in supporting its policy recommendations by better describing the causes of ecosystem conversion historically and by offering more discussion about how these factors could/would be mitigated in the conservation scenario, especially given the conservation scenario assumption of no net Broadleaf forest loss. Also, if there was no significant historic loss of ecosystems due to "Built Areas," I suggest noting this in the paper.

Response: The exclusion of built areas was based on the observation that this class had a negligible contribution to the gain/loss of all other classes between 2007-2015. This has been added in the manuscript now (lines 152-155).